# Hybrid-Learning Type-2 Takagi–Sugeno–Kang Fuzzy Systems for Temperature Estimation in Hot-Rolling

**José Ángel Barrios [1], Gerardo Maximiliano Méndez [2] and Alberto Cavazos [1,***

[1] Posgrado en Ingeniería Eléctrica, Facultad de Ingeniería Mecánica y Eléctrica, Universidad Autónoma de Nuevo León, San Nicolás de los Garza 66455, N.L., Mexico; joseangel_barrios@yahoo.com.mx

[2] Depto. de Ing. Eléctrica y Eléctrónica, Tecnológico Nacional de México, Nuevo León, Guadalupe 67170, N.L., Mexico; gerardo.maximiliano.mendez@gmail.com

* Correspondence: alberto.cavazosgz@gmail.com; Tel.: +52-81-1203-1684

**Abstract:** Entry temperature estimation is a major concern for finishing mill set-up in hot strip mills. Variations in the incoming bar conditions, frequent product changes and measurement uncertainties may cause erroneous estimation, and hence, an incorrect mill set-up causing a faulty bar head-end. In earlier works, several varieties of neuro-fuzzy systems have been tested due to their adaptation capabilities. In order to test the combination of the simplicity offered by Takagi–Sugeno–Kang systems (also known as Sugeno systems) and the modeling power of type-2 fuzzy, in this work, hybrid-learning type-2 Sugeno fuzzy systems are evaluated and compared with the results presented earlier. Systems with both empirically and fuzzy c-means-generated rules as well as purely fuzzy systems and grey-box models are tested. Experimental data were collected from a real-life mill; datasets for rule-generation, training, and validation were randomly drawn. Two of the grey-box models presented here reach 100% of bars with 20 °C or less prediction error, while two of the purely fuzzy systems improved performance with respect to purely fuzzy systems presented elsewhere, however it was only a slight improvement.

**Keywords:** type-2 fuzzy; hot-rolling; temperature estimation; Takagi–Sugeno–Kang

## 1. Introduction

The finishing mill set-up is a crucial issue in hot-rolling as it has to be properly calculated for the bar front section to meet requirements. Some rolling variables, including temperature, at the bar head-end have to be estimated for set-up calculations. Estimation has to be performed online, and quickly, such that the bar maintain as much heat as possible [1,2]. Thus, temperature estimation at the bar head-end is an important concern in hot-rolling.

This work is particularly concerned with the bar-head end temperature estimation at the scale breaker entry, as shown in Figure 1. Physical modeling commonly performs the scale breaker entry temperature estimation in most hot-rolling lines worldwide. Such physical models are based on temperature measurements at the roughing mill exit and the bar traveling time from the roughing mill exit to the scale breaker entry. The modeling is based on the roughing mill exit temperature measurements, since at this point the measurements are cleaner than those at any other subsequent point in the process. In addition, it is not affected by recalescence [3]. The estimation is carried out in cascade according to the different thermal phenomena involved, as shown in Figure 1. Information on how the parameters are taken into account is presented elsewhere [4,5].

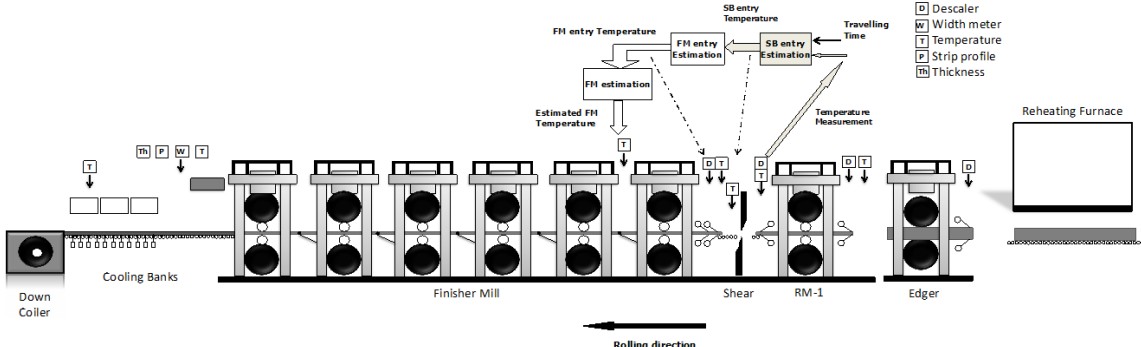

**Figure 1.** Hot-rolling mill production line with the temperature prediction block diagram.

An additive proportional-plus-integral term is often used to compensate the temperature modeling error [6]. The estimation system using the proportional-plus-integral compensation will be referred to as "model + PI" in the rest of this work. In spite of the proportional-plus-integral compensation function, the estimation is greatly affected by process variations, measurement uncertainties, and product changes, i.e., large estimation errors are produced, subsequently resulting in an incorrect mill set-up and a faulty bar head-end [7].

In general, rolling variable estimation is important for control purposes, and hence, for quality fulfilment. Temperature estimation along the strip using physical modelling at different stages of the rolling line have been reported in the literature. A mean error of 0.8° in finishing temperature estimation and 3.9% of coils with an absolute mean error greater than 10° is reported in [8]. Results of 83% of bars with an error lesser than 3° in finishing temperature prediction are presented in [9]. A physical model was implemented on a digital system reporting an error of 15° in the finishing temperature estimation [10].

Computational intelligent techniques, such as artificial neural networks, fuzzy logic, and computational intelligence-based grey-box models (GB); have been widely used in industry due to their capabilities of adaptation [6,11] and human knowledge incorporation [12,13]. A number of applications of computation intelligent techniques on estimation and control, along with GB models, in a wide range of industrial areas such as food processing, iron and steelmaking, chemical, biochemical, pharmaceutical, power plants, oil and gas processing, water treatment, industrial robots, heat treatment, etc., have been recently reported in the literature [14,15].

Due to its importance, research on rolling variables estimation using computational intelligence has also been performed in order to reduce estimation error. Roll thermal expansion prediction in hot-rolling using neural networks has been reported in [16]. In [17], results of a research work to evaluate prediction of bending force by neural networks adapted by a genetic algorithm are reported. The water-cooling heat transfer coefficient is estimated by a Takagi–Sugeno–Kang fuzzy system (also known as a Sugeno system) within a closed-loop control scheme coiling temperature [18].

Fuzzy logic and artificial neural networks have also been tested to overcome the aforementioned temperature estimation problems at the scale breaker entry. One parallel and two series connection structures of neural GB models have been developed and tested [11]. Several factors, which were believed to influence the process, but were not included in the physical model, were also tested, and they were shown to improve prediction for both neural networks and neural-based GBs [6]. Experimental results on adaptive neuro-fuzzy inference systems (ANFISs) and ANFIS-based GB models are presented in the work of Barrios et al. [11]. Type-1 and type-2 Mamdani fuzzy inference systems with hybrid-leaning (HL) adaptation, both, as purely fuzzy and within GB schemes, were also tested for scale breaker entry temperature prediction [11,19]. Fuzzy c-means (FCM)-generated rule-based ANFISs and ANFIS-based GB models were tested and evaluated in [20].

As can be seen from the brief literature review above, computational intelligence techniques have been used to predict the scale breaker entry temperature because of their powerful modeling

capabilities, particularly type-2 fuzzy systems [19]. On the other hand, the Sugeno fuzzy system and its type-1 adaptive version, the so-called ANFIS, have been proven to be successful in a number of applications, despite being simpler than Mamdani fuzzy systems. In this work, HL type-2 Sugeno fuzzy systems, both with empirical and FCM rule generation, are developed and evaluated for scale breaker entry temperature prediction to take advantage of the merits of both techniques combined. Due to the importance of scale breaker entry temperature for the finishing mill set-up, and hence, for bar head-end to meet requirements, the main purpose of this work is to explore the benefits of these two methods combined on scale breaker temperature estimation, given the simplicity of Sugeno systems in relation to Mamdani and the powerfulness of type-2 fuzzy. Both purely fuzzy systems and GB models with 9 and 25 rules are tested. To allow a complete comparison, a HL type-1 Mamdani with FCM-generated rules is also developed here, since reports of the application of such systems have not been found in the literature. They are also developed and tested for both purely fuzzy systems and GB models, with 9 and 25 rules. The benchmark for this work is the physical model, due to the fact that works on bar head-end scale breaker entry temperature estimation, for finishing mill set-up purposes, based on techniques other than those revised and studied in this work, have not been found in the literature to date. Hence, the performance of the systems presented here is compared with that of the model + PI. The systems were designed using MATLAB® (R2019b, The MathWorks Inc., Natick, MA, USA) and tested with data collected from a real hot strip mill. Experimental results show the benefits of applying type-2 Sugeno fuzzy systems and GB models for temperature estimation in a hot strip mill. A list of symbols and acronyms used in this work is given in Table 1.

**Table 1.** List of symbols and acronyms.

| Symbol or Acronym | Meaning |
|---|---|
| Model + PI | Physical model with proportional-plus-integral compensation |
| GB | Grey box |
| FCM | Fuzzy C-means |
| ANFIS | Adaptive neuro-fuzzy inference system |
| HL | Hybrid learning |
| MAE | Mean absolute error |
| RMSE | Root mean squared error |
| %Bars ± 20 °C | Percentage of bars with a prediction error within ±20 °C |
| FOU | Footprint of uncertainty |
| $x_1$ | Entry 1 to fuzzy systems, measured temperature at roughing mill exit |
| $x_2$ | Entry 2 to fuzzy systems, measured traveling time from roughing mill exit to scale breaker entry |
| $o$ | Output of a conceptually defined fuzzy system |
| $w, w_u, w_l$ | Firing strength, $u$ and $l$ denote upper and lower in type-2 fuzzy, respectively |
| $y$ | Estimation of purely fuzzy systems and GB models prediction |
| $z$ | Model + PI estimation error |

For the sake of briefness, the methodology fundamentals are briefly presented in this paper; the reader is referred to [20,21] for FCM fundamentals; fuzzy logic and ANFIS theory can be found in [21]; in [19], a summary of the fundamental principles of type-2 fuzzy logic is presented, while for a deeper insight the reader is referred to [22].

## 2. Materials and Methods

### 2.1. Hot-Rolling Mill

A hot strip mill transforms steel slabs or ingots forms obtained by continuous or traditional casting into a coiled strip. Typical dimensions of the slabs are 10 m long, 1 m wide and 0.2 m thick. A typical hot-rolling line consists of the following stages: furnaces, one or two roughing mills (in the present case there are two roughing mills), a finishing mill, cooling banks and down coilers. Figure 1

shows the hot-rolling line where the present work was undertaken. This hot strip mill is working with a walking beam. A final strip coil of a hot strip mill must attain required thickness, width and mechanical properties [1].

The slabs, originally at ambient temperature, are reheated to around 1300 °C. When a slab reaches the appropriate temperature, it leaves the furnace. The target temperature, and therefore the residence time of every individual slab in the furnace depends on steel grade, slab dimensions and the final product. This is also true for the rolling pace, but it is also determined by every particular mill capability.

After leaving the furnace, the slab is transported for roughing, in this case, by two reversible roughing mills, as shown in Figure 1. Here, the initial thickness reduction takes place usually by 5 or 7 passes. Thickness reduction in the roughing mill is typically from 200 mm to 25.4 mm. The roughing mill output is called a transfer bar, and is typically 90 m in length. The next stage is the finishing mill, which often consists of 6 or 7 stands. In this particular mill there are 6 stands. Once in the finishing mill, the bar is called a strip. At the finishing mill exit, the strip has to fulfil final thickness and width and finishing temperature specifications; the latter is required to achieve the desired mechanical properties. When the strip leaves the finishing mill, it is taken to the cooling banks where the strip has to be cooled down from the finishing temperature to a specific coiling temperature, which is also required for mechanical properties.

An oxide film called the primary oxide is formed over the slab surface during the reheating process in the furnace, and the rolling process when exposed to the environment is called the secondary oxide. The oxide film has to be removed to allow proper rolling. The descaler devices (Figure 1, labelled with the letter D) are equipped with high-pressure water jets in order to remove the oxide layer from the slab surface.

The most critical process in a hot strip mill is the finishing mill. It involves a great number of variables due to the interaction between stands, and it requires a higher level of automation [1,2]. Every stand has to accurately achieve a particular proportion of thickness reduction (draft). The finishing mill also has to fulfil the finishing temperature within a certain tolerance band. In order to obtain a more stable rolling process, a specific strip tension between the slabs is needed. This is supplied by devices called loopers. Tension also contributes to thickness reduction. Therefore, thickness, finishing temperature and tension, among other variables, should be controlled when the strip is being rolled within the finishing mill. However, the controllers' set points are not straight forwardly obtained since the incoming bar conditions, such as temperature and resistance, may vary from bar to bar. The specifications of the final product, which for the finishing mill are final thickness and width and finishing temperature, may also change. Therefore, the set points for controllers have to be calculated and must be sent before the incoming bar enters the finishing mill, which is crucial in order for the front section of the coil to meet the specifications.

*2.2. Methogology Fundamentals*

2.2.1. Fuzzy Logic

Fuzzy logic is essentially a multiestimated logic that is an extension of classical logic. The latter uses only the terms "true" and "false" and assigns deterministic values to its variable. This logic satisfactorily models a great part of the "natural" reasoning. Although human reasoning uses "true" or "false" values, they are not necessarily "so deterministic". Fuzzy logic intends to produce exact results from vague information, which are particularly useful in electronic or computational applications. The "fuzzy" adjective refers to the nondeterministic values that, in general, have an uncertain connotation.

The two kinds of fuzzy systems more commonly used are Mamdani and Sugeno. In both cases a set of rules of the form "IF-THEN" are used to model the problem. A fuzzy rule has the following form:

Mamdani: if $x_{1i}$ is $A_{1i}$ and $x_{2i}$ is $A_{2i}$ then $o_i$ is $A_{oi}$

Sugeno: if $x_{1i}$ is $A_{1i}$ and $x_{2i}$ is $A_{2i}$ then $o_i = f(x_{1k}, x_{2k})$

where $x_1$ and $x_2$ are input variables; $o$ is the output variable; $A_{ji}$ is the fuzzy set for input $j$, and $A_{oi}$ is the output fuzzy set defined within the operating ranges of $x_1$, $x_2$ and $o$ respectively; $f(x_1, x_2)$ is a linear function; and $i$ is rule number. A fuzzy set is commonly described by a triangular, trapezoidal or a Gaussian function, which is a called membership function since it gives the degree of membership of a particular entry dataset to a given fuzzy set.

A rule expresses the relation between the input fuzzy sets $A_{1i}$ and $A_{2i}$ and the output fuzzy set $A_{oi}$, whose typical function would be $\mu_{A1i \wedge A2i \rightarrow Aoi}$, where the operator $\wedge$ denotes the AND operation, which in fuzzy set operations, denotes an intersection implemented as a minimum operation. This operation is called the t-norm. Anther commonly used function is $\mu_{A1i \vee A2i \rightarrow Aoi}$, where the operator $\vee$ denotes the OR operation, which in fuzzy sets operations, represents a union operation implemented as a maximum operation. This operator is called the t-conorm. These operations represent—and are known as—logical implications. The input part of the rule is called the antecedent and the output part, preceded by the preposition "then", is known as the consequent. Figure 2a show a logical implication mechanism using the t-norm operation, which in this work, is used for the antecedent.

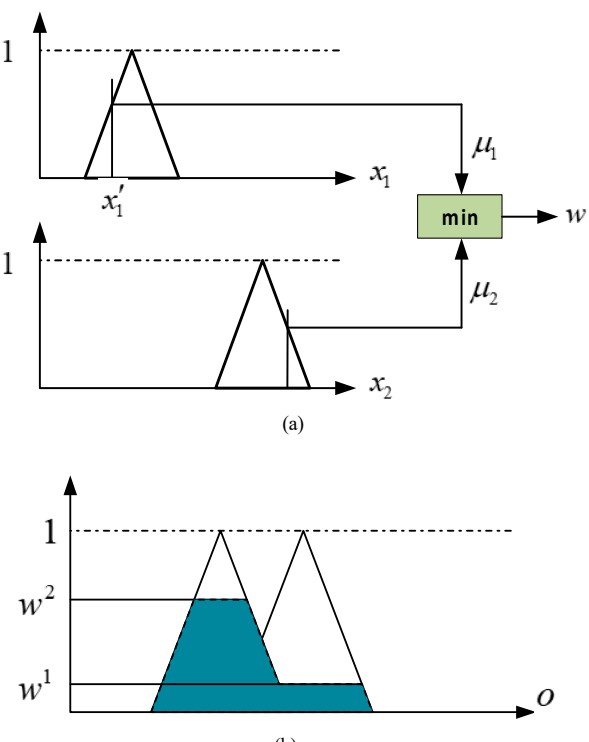

(a)

(b)

**Figure 2.** Type-1 fuzzy inference mechanism: (**a**) antecedent logical implication and (**b**) aggregated consequent set for two-rule evaluation.

In a Mamdani type fuzzy systems, the consequent is a fuzzy statement (see rules above); therefore, the output from each rule is a fuzzy set produced from the projection of the firing strength ($w$) over the output fuzzy set. Since the t-conorm is assumed in Figure 2b for the consequent operation, an aggregated fuzzy set is produced from a logical operation between each rule output for a given input dataset. Nonetheless, in engineering, a single value is needed, therefore a defuzzification step is carried out. There are commonly four methods for defuzzification: centroid of area, bisector of area, mean of maximum, and smallest of maximum. In this work the centroid of area is used, and it is given by

$$o_{df} = \frac{\int \mu_{Ao}(o)o\,do}{\int \mu_{Ao}(o)\,do} \tag{1}$$

where $\mu_{Ao}(o)$ is the aggregated output fuzzy set and $o_{df}$ is the defuzzified output.

The consequent in a Sugeno type fuzzy system is a deterministic function (see rules above); therefore, in this work, the zero order Sugeno system is used with the output given by

$$o_i = \frac{\sum_i w_i p_i}{\sum_i w_i} \tag{2}$$

where $p_i$ is a constant.

The type-2 fuzzy membership functions are as above for type-1 fuzzy (although with uncertain means) for both antecedents ($x_1$ and $x_2$) and consequents ($o$). In this work, it is assumed that the interval type-2 membership function case holds [19,22].

Then, the type-2 membership function with uncertain means for each antecedent is expressed as

$$\mu_i(g_l) = \exp\left[-\frac{1}{2}\left[\frac{g_l - m_i^{Ai}}{\sigma_i}\right]\right] \tag{3}$$

where $m_{in} \in [m_{i1}, m_{i2}]$ is the uncertain mean; $n = 1, 2$ is the lower and upper bounds of the uncertain mean; $l = 1, 2$ is the input number; and $\sigma_i$ is the standard deviation. The uncertain mean forms the so-called footprint of uncertainty (FOU) which is depicted in Figure 3a in a type-2 fuzzy logical implication.

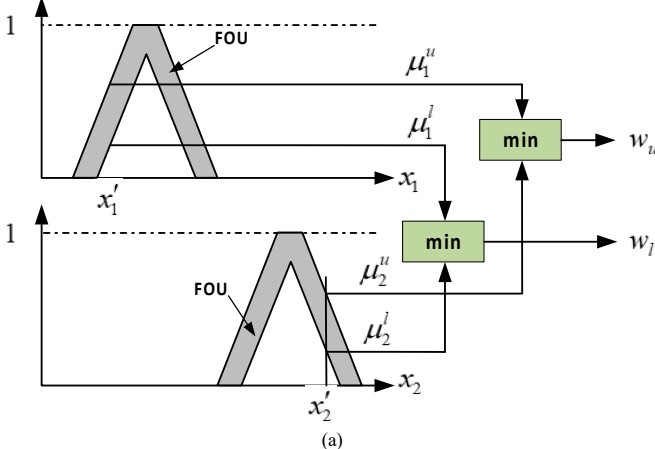

(a)

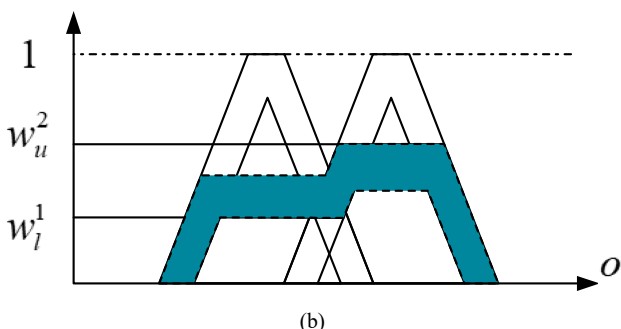

(b)

**Figure 3.** Type-2 fuzzy inference mechanism: (**a**) antecedent logical implication and (**b**) aggregated consequent set for two-rule evaluation.

The type-2 fuzzy consequent logical implication is shown in Figure 3b, as well as an aggregated fuzzy set using upper and lower firing strengths. The defuzzification process is considerably more cumbersome using the iterative algorithm given in [19,22].

### 2.2.2. Fuzzy C-Means

K-means clustering (also called C-means clustering) is a data-grouping algorithm which partitions input–output data pairs into groups of clusters according to some similar characteristics, usually given as an objective function which has to be minimised. Such clustering algorithms are commonly used to determine the initial set of rules of a fuzzy system. A collection of $n$ vectors $x_j$, $j = 1, 2, 3, \ldots n$, is partitioned into $c$ groups $c_i$, $i = 1, 2, 3, c$; $c$ cluster centres are then determined by minimizing the following cost function:

$$J = \sum_{i=1}^{c} J_i = \sum_{i=1}^{c} \left( \sum_{k, x_k \in C_i} \|\mathbf{x}_k - \mathbf{c}_i\|^2 \right) \tag{4}$$

where $J_i = \sum_{k, x_k \in C_i} \|\mathbf{x}_k - \mathbf{c}_i\|^2$ is the cost function for group $i$ and $c_i$ is the centre of group $i$. In general, any similarity function may be used for $J_i$, however, the Euclidean distance is the most commonly used. In K-means clustering, a particular data vector ($\mathbf{x}_k$) only belongs to the group with to the closest centre $c_i$.

A membership matrix U is formed as follows:

$$u_{ij} = \begin{cases} 1 \; if \; \|\mathbf{x}_j - \mathbf{c}_i\|^2 < \|\mathbf{x}_j - \mathbf{c}_k\|^2, for \; all \; k \neq i \\ 0 \; otherwise \end{cases} \tag{5}$$

In order to improve this algorithm, fuzzy C-means clustering (FCM) was proposed. The main difference is that in fuzzy C-means, $\mathbf{x}_k$ may belong to more than one group according to a degree of membership. The total degree of membership for a particular data vector ($\mathbf{x}_k$) should equal unity.

### 2.3. Experimental Data

Since the bar arrival time is unknown, prediction is updated every 5 s, which subsequently produces an inherent prediction error. Moreover, as mentioned, the scale breaker entry temperature measurement is not as reliable as that at the roughing mill exit. Therefore, the so-called reprediction is performed after the arrival time is collected and the scale breaker entry temperature measurement has been validated. The model + PI compensation is performed based on the repredicted temperature, as shown in the estimation flow-chart in Barrios et al. [6]. Data of 42,000 consecutive bars were collected from a real hot strip mill, and the date collected, among others, were the roughing mill exit time, the roughing mill exit measured temperature, the scale-breaker bar arrival time, the scale breaker entry temperature, and the model + PI repredicted temperature. Data of 37,000 bars out of the 42,000 originally collected were kept after inconsistencies were removed. Three different sets were randomly drawn, the first one, consisting of 10,000 data vectors, was used to generate the rule-bases of the FCM algorithm, while the other two sets of 3700 data vectors each were used for training and validation of the fuzzy systems. The ranges of $x_1$, $x_2$ and $y$ are 1072–1192 °C, 16.5–20.5 s, and 1045–1125 °C respectively.

### 2.4. Fuzzy Systems for Scale Breaker Entry Temperature Estimation

The inputs to the physical model to estimate the scale breaker entry temperature ($y$) are the surface temperature measured at the roughing mill exit ($x_1$) and the bar travelling time from the roughing mill exit to the scale breaker entry ($x_2$). These are the inputs to the purely fuzzy systems and GB models developed here to estimate the scale breaker entry temperature, as shown in the shaded block in Figure 1. In practice, the travelling time is estimated, and it is recursively updated while waiting for bar arrival. Nevertheless, as mentioned, the measured travelling time after bar arrival is used to compensate the model estimation for the next bar; therefore; in this work, the measured travelling time is used.

### 2.5. Type-2 Sugeno Fuzzy Systems and GB Models for Scale Breaker Entry Temperature Modeling

As mentioned, the systems are tested as purely fuzzy systems or within GB models for 9 or 25 rules; Gaussian functions are used as membership functions as in previous works. The performance of the type-2 Sugeno purely fuzzy and GB models is evaluated and compared with the systems developed in the earlier work and those not found in the abovementioned literature. In this way, a complete study of neuro-fuzzy systems applied to scale breaker entry temperature prediction in an hot strip mill is presented.

In this work, parallel GB models are used. These are compound structures combining two or more models of different nature. One of those systems is a fuzzy system, and the other one is the model + PI. The fuzzy system output is an additive term. The merits and justification of using GB models can be found in [11]. The fuzzy systems within the GB structures (as shown in Figure 4) are also designed using the methodology for the fuzzy systems, with either 9 or 25 rules as described in earlier works; however, the output data vector is given by the model + PI estimation error (instead of $y$), i.e., the fuzzy system will predict the model + PI error, such that it may compensate for it. The fuzzy system in Figure 4 has the same inputs, $x_1$ and $x_2$, while the output is the fuzzy system prediction of the model + PI estimation error ($z$). Note that the output of the GB model is still called $y$, while in Figure 4, the model + PI estimation is called $\hat{T}_m$. The range of $z$ is −75–25 °C [19].

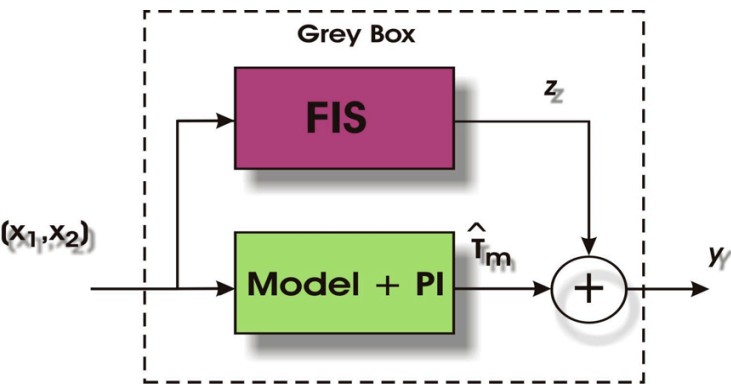

**Figure 4.** Parallel Grey-box scheme.

In the following section, the type-2 Sugeno with FCM rule generation designed is described (for type-1 and type-2 Mamdani, see [11,19,20]). The fuzzy systems developed here have rules in the form of those given in Section 2.2.1, where $x_{1k}$, $x_{2k}$ and $o_k$ are the measured temperature at the roughing mill exit; the measured bar travelling time; and the fuzzy prediction output, either, temperature at scale breaker entry ($y$) or the model + PI prediction error ($z$); for the $k$th data pair, respectively.

The FCM method is a clustering algorithm, which partitions the input/output data into groups, supplying the group centroid ($c_i$) and standard deviation ($\sigma_i$). As mentioned, the fuzzy sets ($A_{ji}$ and $A_{oi}$) are described by Gaussian membership functions distributed according to $c_i$ calculated by the FCM algorithm, which determines the Gaussian membership function mean values ($u_i$). $\sigma_i$ determines the standard deviation of the $i$th membership function and $i$ denotes the cluster number corresponding to cluster-$i$ and rule number. The fuzzy rules relate the corresponding input/output clusters; thus, instead of having rules for all membership function combinations of fuzzy sets as done with the empirical rule basis [11,19], there is only one rule per cluster. To establish the type-2 membership functions, a noisy mean is assumed, bounded by upper and lower limits denoted here as $u_i{}^l$ and $u_i{}^u$ respectively, assuming constant $\sigma_i$.

The FCM algorithm was run for 100 epochs using the 10,000 data vector set randomly drawn from the data collected as described above. Once the data clusters were generated, the fuzzy systems were designed and trained using the randomly drawn training data during the 12-epoch period.

After training, the system performance is evaluated with the validation dataset. This procedure is iterative, that is, it varies the number of epochs until satisfactory results are obtained.

## 3. Results

The FCM algorithm was run to group the data and the noisy mean was applied on the mean values obtained to generate the type-2 fuzzy sets. Figures 5 and 6 show the type-2 membership functions for the 9-rule purely fuzzy systems and GB models respectively, and the outcome of the FCM algorithm for both can be found in [20].

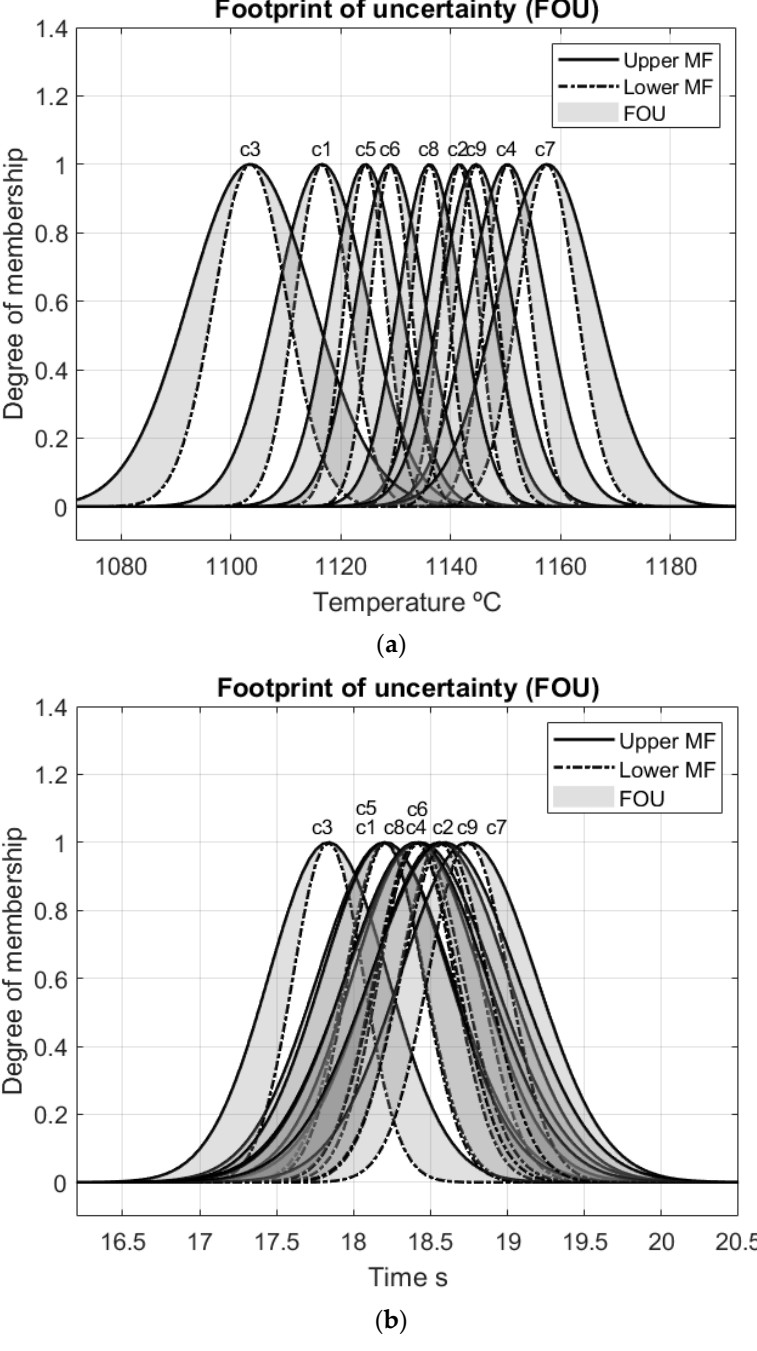

(**a**)

(**b**)

**Figure 5.** *Cont.*

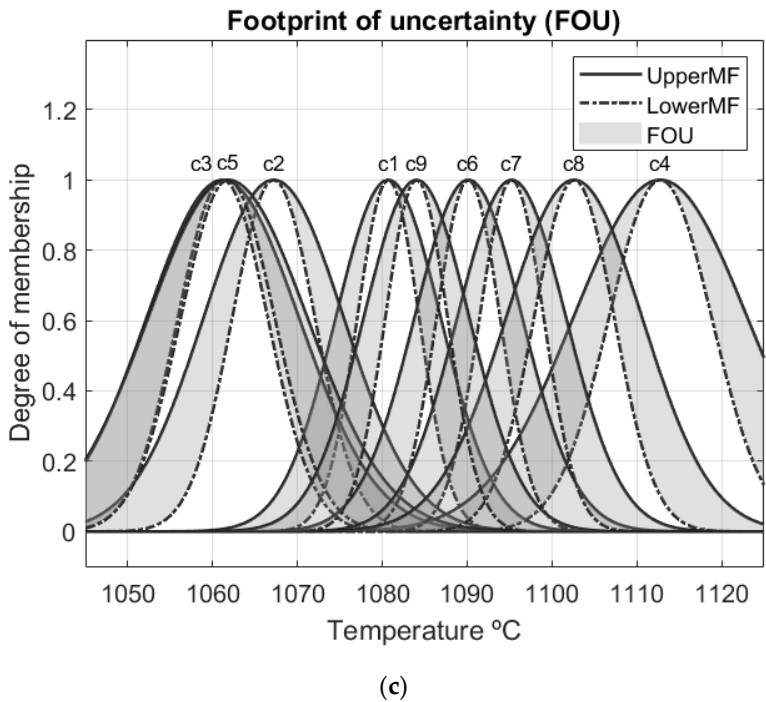

(**c**)

**Figure 5.** Fuzzy C-means (FCM) 9-rule Mamdani purely fuzzy system input and output type-2 membership functions, (**a**) $x_1$, (**b**) $x_2$ and (**c**) $y$.

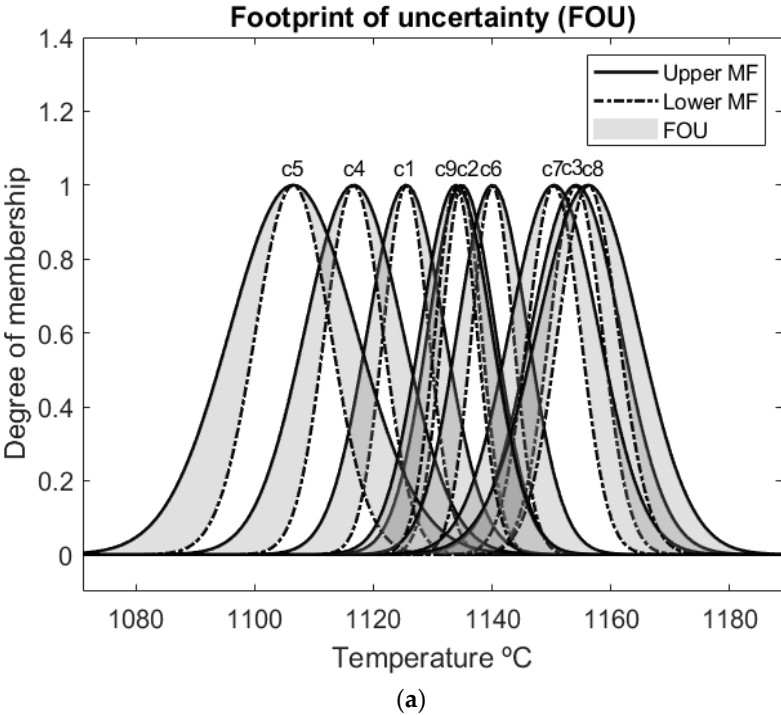

(**a**)

**Figure 6.** *Cont.*

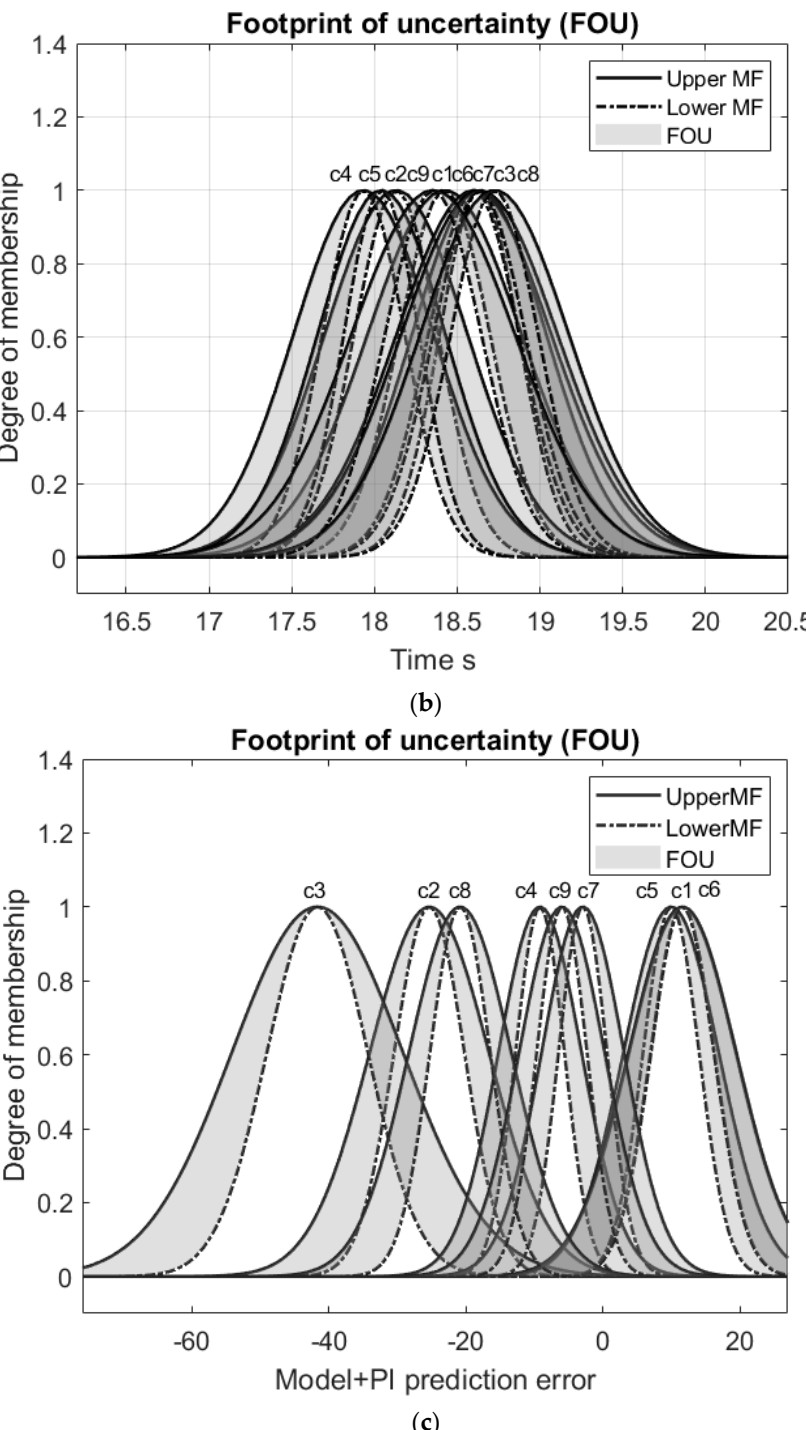

**Figure 6.** FCM 9-rule Mamdani GB input and output type-2 membership functions, (**a**) $x_1$, (**b**) $x_2$ and (**c**) $z$.

Note that for the Sugeno fuzzy system, there are not output membership functions. The ones shown correspond to Mamdani type-2 fuzzy system with FCM rule generation. In the type-2 Sugeno fuzzy system, the output level $v_i$ is calculated as follows:

$$v_i = \sum_{j=1}^{m} p_j^i x_j^i \qquad (6)$$

where $v_i$ is the output level for the *i*th rule, $x_j$ is the actual measurement of *j*th input, *m* is the number of inputs (2) and $p^i{}_j$ is a consequent parameter, which in this case, is the mean value of the membership function from the Mamdani fuzzy system corresponding to $y_i$ or $z_i$. In type-2 Sugeno, the aggregated set is a function of $v_i$ and the lower and upper values of the firing strength ($w_l$ and $w_u$). An aggregated set for a particular estimation is shown in Figure 7. The type reduction method to obtain the output (*y* or *z*) is the same as the type-2 Mamdani fuzzy system and it can be found in the reference provided above.

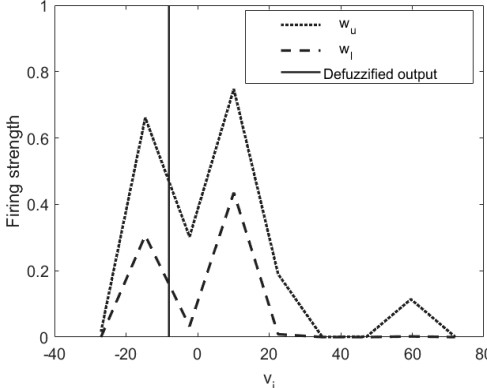

**Figure 7.** Type-2 Sugeno fuzzy output aggregated set for a particular estimation.

Five performance indices were used to evaluate the system performance. These performance indices were applied over the estimation error with the validation set as in the previous works and described above. The estimation error is defined as

$$e = T_e - T_m \tag{7}$$

where $T_e$ is the temperature estimated by the particular system to be evaluated and $T_m$ is the measured temperature at the scale breaker entry area. The performance indices are: (1) mean error; (2) standard deviation; (3) mean absolute error (MAE); (4) root mean square error (RMSE); and (5) percentage of bars of the validation dataset with an estimation error within ±20 °C, abbreviated as '%Bars ± 20 °C'. In practice, there are no standard specification limits for the entry temperature estimation error in the finishing mill; however, such a performance index would be a very illustrative indicator for evaluation purposes [6,8,18,19]; ±20 °C was found to be a suitable tolerance, details are given in [6].

When evaluating the systems, a mean error closer to zero is pursued, while for the standard deviation, MAE and RMSE, low values should be expected. A large '%Bars ± 20 °C' is desirable.

Table 2 shows only the '%Bars ± 20 °C' performance index of the different varieties of fuzzy systems tested so far, since it allows a more straight-forward comparison. The purely fuzzy systems and GB models developed in this work are highlighted with shaded boxes. Bold characters indicate the best purely fuzzy system and the best GB model in terms of '%Bars ± 20 °C'. Note that Sugeno type-1 fuzzy systems with HL are the so-called ANFIS. Here they are denoted as HL Sugeno type-1 fuzzy in order to identify them among the variety of fuzzy systems presented.

**Table 2.** '%Bars ± 20 °C' of the fuzzy systems tested.

| System | Rules | Untrained | FCM | HL | HL/FCM | GB | GB/FCM | GB/HL | GB/HL/FCM |
|---|---|---|---|---|---|---|---|---|---|
| Type-1 Sugeno | 9 | 77.65 | 77.94 | 79.32 | 78.45 | 55.55 | 95.95 | 99.66 | 98.98 |
| Type-1 Mamdani | 9 | 77.41 | 76.48 | 77.32 | 77.43 | 56.56 | 88.55 | 100 | 99.07 |
| Type-1 Sugeno | 25 | 68.03 | 73.91 | 78.40 | 78.05 | 80.80 | 86.86 | 95.62 | 100 |
| Type-1 Mamdani | 25 | 68.21 | 71.75 | 78.78 | 79.32 | 80.80 | 75.08 | 99.32 | 98.10 |
| Type-2 Sugeno | 9 | 76.67 | 78.11 | 79.78 | 78.29 | 53.87 | 95.62 | 95.95 | 100 |
| Type-2 Mamdani | 9 | 77.32 | 76.10 | 78.43 | 78.54 | 49.32 | 87.67 | 99.51 | 99.40 |
| Type-2 Sugeno | 25 | 68.35 | 78.11 | 80.11 | 79.08 | 80.81 | 91.24 | 97.04 | 100 |
| Type-2 Mamdani | 25 | 75.89 | 74.59 | 79.27 | 79.02 | 73.70 | 89.27 | 99.35 | 97.02 |

In order to allow for a more detailed comparison, Table 3 shows the performance indices of the top four GBs, with 100% of '%Bars ± 20 °C', ranked by mean error.

**Table 3.** Performance indices of the GB models with 100% of '%Bars ± 20 °C' ranked by mean error.

| Ranking | Variety | No. of Rules | Standard Deviation | Mean Error | MAE | RMSE | %Bars ± 20 °C |
|---------|---------|--------------|--------------------|------------|-----|------|----------------|
| 1 | HL type-1 Mamdani | 9 | 5.35 | −0.17 | 4.16 | 5.35 | 100 |
| 2 | HL/FCM type-2 Sugeno | 25 | 6.57 | 0.78 | 5.25 | 6.61 | 100 |
| 3 | HL/FCM type-2 Sugeno | 9 | 5.78 | 3.39 | 4.72 | 6.7 | 100 |
| 4 | HL/FCM type-1 Sugeno | 25 | 4.45 | 5.42 | 5.63 | 7.02 | 100 |
| 5 | Model + PI | n/a | 17.35 | −6.14 | 14.44 | 18.40 | 73.97 |

In the same way, Table 4 shows the performance indices of the top four purely fuzzy systems.

**Table 4.** Performance indices of the top four purely fuzzy systems ranked by '%Bars ± 20 °C'.

| Ranking | Variety | No. of Rules | Standard Deviation | Mean Error | MAE | RMSE | %Bars ± 20 °C |
|---------|---------|--------------|--------------------|------------|-----|------|----------------|
| 1 | HL type-2 Sugeno | 25 | 15.49 | −0.54 | 12.61 | 15.5 | 80.11 |
| 2 | HL type-2 Sugeno | 9 | 15.57 | −0.55 | 12.66 | 15.58 | 79.78 |
| 3 | HL/FCM type-1 Mamdani | 25 | 15.65 | −2.3 | 12.75 | 15.82 | 79.32 |
| 4 | HL type-1 Sugeno | 9 | 15.57 | −3.34 | 12.76 | 15.93 | 79.32 |

Scatter plots show predictions against measurements; hence, the ideal prediction would be a unitary ramp. In this way, the prediction dispersion of a particular system is readily appreciated, allowing a visual comparison between deferent systems. Scatter plots of the temperature predictions for the two Sugeno type-2 fuzzy GB models in Table 3, namely, the 25-rule HL/FCM Sugeno type-2 GB model and the 9-rule HL/FCM type-2 Sugeno GB model, are shown in Figures 8 and 9, respectively. The ideal prediction line is the one plotted in white color.

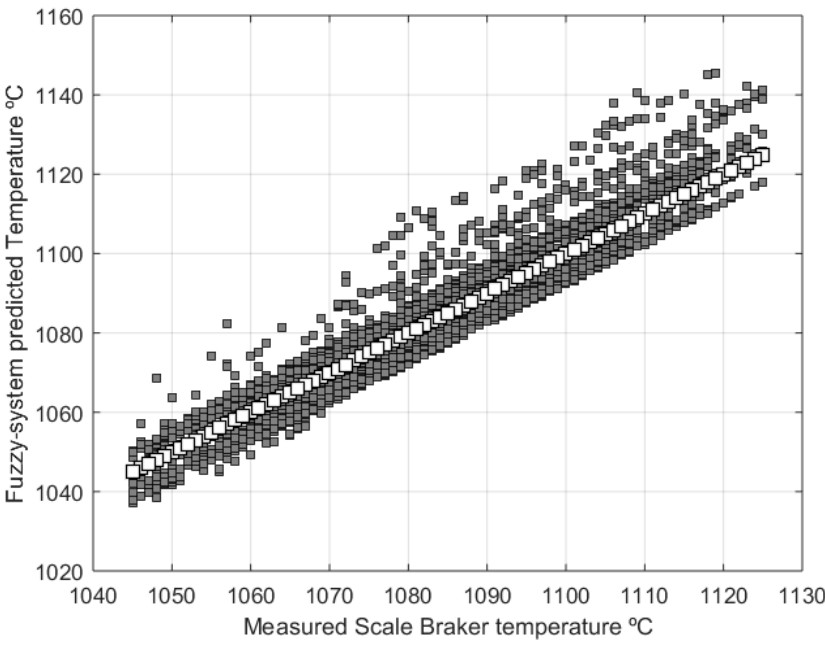

**Figure 8.** Scatter diagram of the 25-rule HL/FCM type-2 Sugeno GB model predicted scale breaker entry temperature.

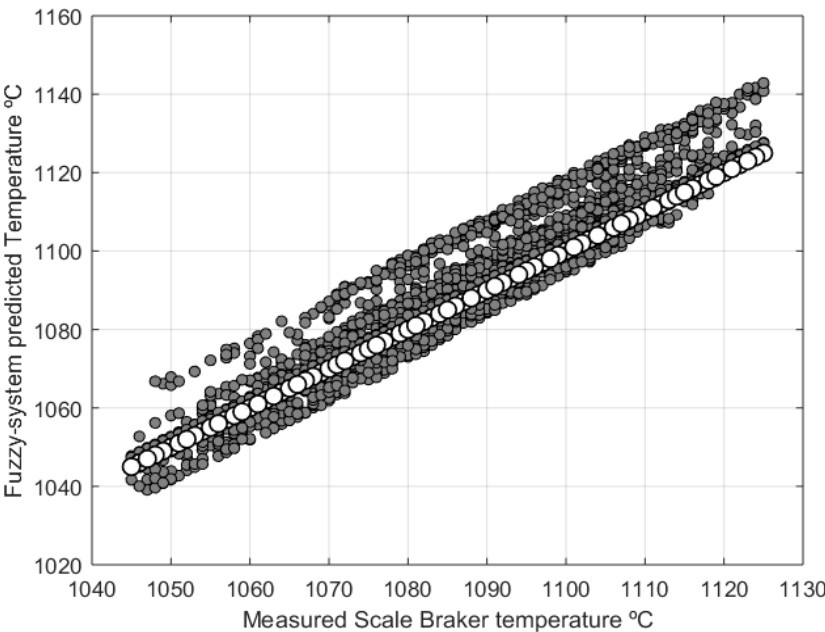

**Figure 9.** Scatter diagram of the 9-rule HL/FCM type-2 Sugeno GB model predicted scale breaker entry temperature.

Similarly, scatter plots of the two Sugeno type-2 purely fuzzy systems in Table 4, namely, the 25 rule HL type-2 Sugeno purely fuzzy system and the 9-rule HL type-2 Sugeno purely fuzzy system, are shown in Figures 10 and 11, respectively. As in Figures 8 and 9, the ideal prediction line is the one plotted in white color.

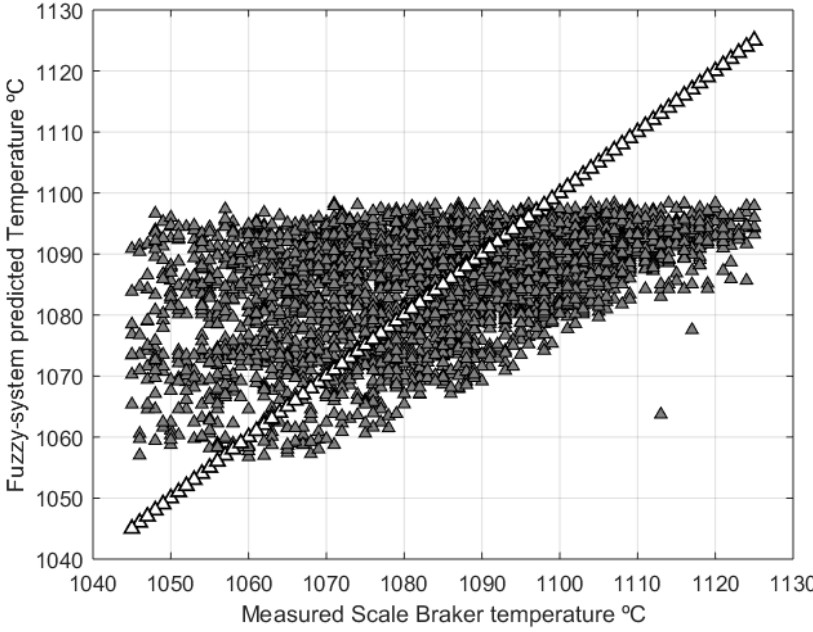

**Figure 10.** Scatter diagram of the 25-rule HL type-2 Sugeno purely fuzzy system predicted scale breaker entry temperature.

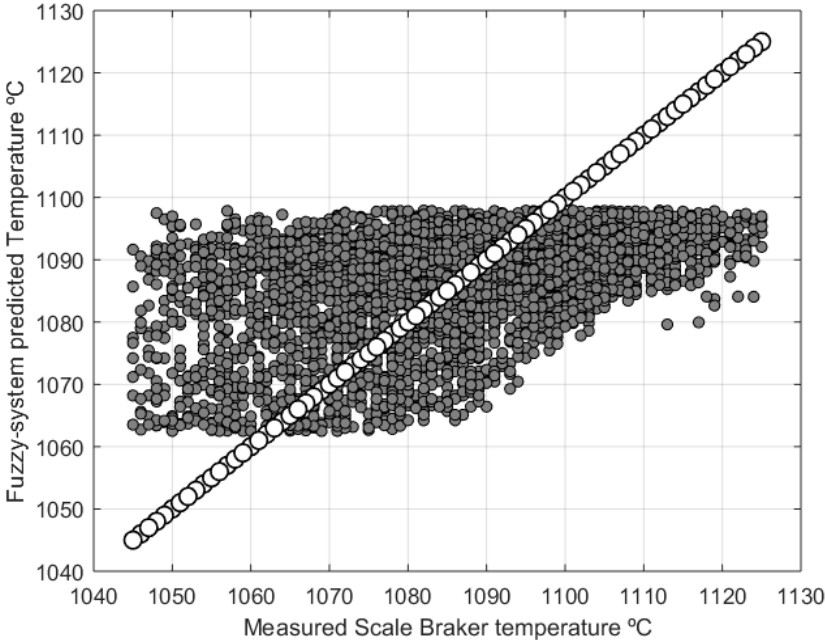

**Figure 11.** Scatter diagram of the 9-rule HL type-2 Sugeno purely fuzzy system predicted scale breaker entry temperature.

## 4. Discussion

As can be seen in Table 2, GB models have a considerably better performance than purely fuzzy systems, as would be expected, since the GBs imply a more complex modeling, demanding more computing resources. As found in earlier works, GBs have a tendency to overpredict, as shown in the mean error in Table 3, but this was not true for the 9-rule HL type-1 GB Mamdani [20]. GBs compensate for the model + PI estimation error, which tends to underpredict (Table 3); hence, the GBs seem to overcompensate.

Table 3 shows the performance indices of the top four GBs, with 100% of '%Bars ± 20 °C', ranked by mean error. It also shows the model + PI performance indices for reference. As can be seen, their standard deviations are similar, and as a result, they have similar dispersion. The same can be said for MAE and RMSE. On the other hand, MAE and RMSE also show similar values, meaning that there are not relatively large errors.

Table 4 shows the performance indices of the top four purely fuzzy systems. Purely fuzzy systems have much smaller mean values than GBs (Table 3); however, their standard deviations, and hence their dispersion, are higher than those of GB models—about three time as high. MAE shows that there are larger magnitude errors than in the case of GBs, but since RMSE is not much greater than MAE, there are not particularly large errors either.

As can be seen in Table 2, the best two purely fuzzy systems are the Sugeno type-2 fuzzy systems for the data tested here. Although there is not a particular improvement when applying type-2 fuzzy systems when compared to type-1 fuzzy systems in general, the type-2 Sugeno system with 25 rules is the only purely fuzzy system with a performance above 80% in terms of '%Bars ± 20 °C'. However, there is a tendency for type-2 fuzzy systems to make the application of a combination of HL and FCM for GBs more favorable—particularly for the Sugeno system. In the other varieties of GBs, HL alone brought larger benefits in five out of eight GBs when compared with HL/FCM.

HL, FCM and HL/FCM improve performance in general. GB models alone, without HL and/or FCM, only bring benefits with respect to purely fuzzy systems when applying 25 rules, suggesting that 9 rules are insufficient to estimate the model + PI prediction error behavior. FCM alone brings

improvements in purely fuzzy systems (except in three cases) but as mentioned, HL alone is generally superior to FCM.

Systems without training exhibit larger difference between MAE and RMSE (not shown here), indicating some relatively large errors. As can be seen from Tables 2 and 3, the model + PI outperforms some of the fuzzy-based systems with no HL nor FCM in terms of '%Bars ± 20 °C', in fact, the systems with the worst performance are 9-rule GB models with no HL nor FCM, reinforcing the abovementioned suggestion that the model + PI prediction error is not easily modelled by a small empirical rule-base.

Scatter plots of the temperature predictions for the two type-2 fuzzy GB models in Table 3, the 25-rule HL/FCM Sugeno type-2 GB model and the 9-rule HL/FCM type-2 Sugeno GB model, are shown in Figures 8 and 9, respectively; while the top two purely fuzzy systems of Table 4, the 25 rule HL type-2 Sugeno purely fuzzy system and the 9-rule HL type-2 Sugeno purely fuzzy system, are shown in Figures 10 and 11, respectively. The results presented in Tables 2–4 can be graphically appreciated in these figures. Figures 8 and 9 show that predictions of the GB models tend to be above the trend line, showing an overprediction tendency as concluded above. On the other hand, comparing both figures, it can also be noticed that the 25-rule HL/FCM Sugeno type-2 GB model is more disperse than the 9-rule HL/FCM type-2 Sugeno GB model, although the former is closer to the trend line. As can be seen in Figures 10 and 11, predictions of the purely fuzzy systems are poor and they show a peculiar behavior of laying all predictions below a straight line just under 1100 °C. A similar behavior can be found in [6] for an artificial neural network-based GB model, although that work was performed with data collected under different temperature conditions; clearly, predictions have to be improved. It is also evident from Figures 8–11 that the GB model predictions are closer to the trend line and less disperse than those of the purely fuzzy systems. Scatter diagrams of the model + PI can be found in [19], and it can be seen that its predictions are more disperse than those of the fuzzy systems shown in Figures 8–11.

As mentioned, two out of the four GB model with 100% of '%Bars ± 20 °C' and the top two purely fuzzy systems are HL type-2 Sugeno systems, however the benefits brought to the purely fuzzy systems are marginal with respect to type-1 Mamdani. The ultimate goal is to implement the kind of systems presented here in the real world, and thus, this work may be useful as a guideline if this were to be the case. In order to make a choice, due to the similar performances shown here, evaluation of the algorithms' efficiency in terms of computing time should be performed before the actual real hot strip mill implementation. It would depend on the computing platform used, as well as the particular algorithm implementation.

Grey-box models have considerably better performance than purely fuzzy systems, however, a physical model is not always available; therefore, further studies to improve purely fuzzy system estimations should be pursued in future, particularly estimations of those showing the best performance in this work, as shown in Table 4.

## 5. Conclusions

In this work, HL type-2 Sugeno fuzzy systems, with rule base generated empirically and by a fuzzy c-means algorithm, have been tested for temperature estimation in hot-rolling. The performance of these systems has been evaluated and compared with that of several varieties of fuzzy systems presented in previous works. In general, 64 systems were considered here with the following varieties: type-1 or type-2 fuzzy, 9 or 25 rules, purely fuzzy or grey-box models, with or without hybrid learning, with or without FCM rule generation, and combinations of hybrid-learning and FCM. The systems that exhibited the best performance, that is, 100% in terms of '%Bars ± 20 °C', were four grey-box models, two of them being type-2 Sugeno fuzzy systems. One of these grey-box models used hybrid-learning and the other three used the combination of fuzzy c-means and hybrid-learning. Although purely fuzzy systems have better mean prediction error, the rest of their performance indices are poorer than those of the grey boxes. Hence, the systems with better performance are the more complex ones, combining all the modeling tools tested here, i.e., GB, HL, and FCM. From these results, it is evident

that a complex design modeling yields better results, as expected. Although, this is at the expense of computing resources. On the other hand, the top two purely fuzzy systems are also type-2 Sugeno fuzzy. It is important to undertake further studies to improve purely fuzzy system estimation, since a physical model is not always available.

**Author Contributions:** Conceptualization, J.Á.B., G.M.M. and A.C.; methodology, J.Á.B., G.M.M. and A.C.; software, J.Á.B., and G.M.M.; validation, J.Á.B., G.M.M. and A.C.; formal analysis, J.Á.B., G.M.M. and A.C.; investigation, J.Á.B., G.M.M. and A.C.; resources, A.C.; writing—original draft preparation, A.C.; funding acquisition, A.C. All authors have read and agreed to the published version of the manuscript.

**Funding:** This research was funded by Universidad Autónoma de Nuevo León, grant number IT394-15.

**Acknowledgments:** The authors acknowledge the Consejo Nacional de Ciencia y Tecnología (CONACYT). The authors would like to thank Luis Leduc and Jorge Ramírez from TERNIUM-Hylsa.

**Conflicts of Interest:** The authors declare no conflict of interest.

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
