# Peer review of "Hybrid-Learning Type-2 Takagi–Sugeno–Kang Fuzzy Systems for Temperature Estimation in Hot-Rolling"

_metals, doi:10.3390/met10060758_

Round 1
Reviewer 1 Report
The authors present a comparison between several models for predicting the entry temperature of a hot rolling mill. Below are several aspects that these authors should improve in their paper:
- Self-references are used in excess in the paper. Of the total number of references included in the paper bibliography, approximately half are the work of any of the authors.
- A more extensive description of the rolling mill on which the data was collected is required. The same for the subsequent treatment of the experimental data.
- The Results section is very brief and does not really present any results. It only indicates how the error of the models will be evaluated, but the results are shown in the Discussion section. Both sections should be reorganized.
- In order to facilitate the reading and understanding of the tables and figures in which the authors present their results, they should be explained in greater detail and properly ordered, facilitating comparison. For example, Figures 6 to 9 appear consecutively without any explanatory text.
- Authors should compare their results with those of other researchers to improve the content of the Discussion section.
Author Response
Dear Reviewers,
We are thankful for your time and valuable recommendations, which greatly contribute to the improvement of the article. The amendments addressing the Reviewer 1 suggestions are written in blue characters in the main text, while those of Reviewer 2 are in green and the answers to Reviewer 3 are in red characters. In the case where two o more reviewers suggested similar changes, these are in orange characters.
Reviewer 1
Comments and Suggestions for Authors
Dear reviewer, we appreciate your time and your recommendations. The amendments to your observations are in blue characters in the main text or in orange characters, if similar recommendations were done by another reviewer.
The authors present a comparison between several models for predicting the entry temperature of a hot rolling mill. Below are several aspects that these authors should improve in their paper:
- Self-references are used in excess in the paper. Of the total number of references included in the paper bibliography, approximately half are the work of any of the authors.
Response
Thank you for this observation, we removed four of our self-references, making indirect citations, and included eight new ones, only one of them is from 2009 and the rest are within the last 5 years. We organize them in three groups as follows:
- Unfortunately, we did not find works directly related to the ours, i.e. scale breaker entry temperature estimation, we included three research works in which temperature at different points of the hot rolling line is estimated by physical modeling, including scale breaker, however evaluated by the performance to estimate the finishing temperature. In fact, the only previous work related to the ours, that we have found, is the physical model used at the rolling mills with which we are comparing.
- We included two works with the aim of showing that estimation and control in industry applications via computational intelligence techniques is a research active area.
- We included three works in which some rolling variable are estimated by intelligent techniques in hot rolling, showing that variable estimation by intelligent techniques is an active research area in hot rolling.
The changes are in orange color characters since another reviewer made a similar suggestion.
- A more extensive description of the rolling mill on which the data was collected is required. The same for the subsequent treatment of the experimental data.
Response
Thank you for the observation. We included a section in blue character describing the mill. However, we do not make much subsequent treatment of the data, apart from removing inconsistencies, drawing the three random sets that we need for grouping, learning and validation and process it with the fuzzy systems. The collection system already filtered it and has an algorithm to validate it. However, we included the following in the experimental data section 2.3:
Since the bar arrival time is unknown, prediction is updated every 5s, producing this an inherent prediction error. Besides, as mentioned, the scale breaker entry temperature measurement is not as reliable as that at the roughing mill exit. Therefore, the so-called re-prediction is performed after the arrival time is collected and the scale breaker entry temperature measurement has been validated. The model+PI compensation is performed based on the re-predicted temperature, see the estimation flow-chart in Barrios et al. [6]. Data of 42000 consecutive bars were collected from a real hot strip mill, the date collected among other were roughing mill exit time, roughing mill exit measured temperature, scale breaker bar arrival time, the scale breaker entry measured temperature, and the model+PI re-predicted temperature. Data of 37000 bars out of the 42000 originally collected were kept after inconsistencies were removed. Three different sets were randomly drawn, the first one, consisting of 10000 data vectors, was used to generate the rule-bases by the FCM algorithm, while the other two sets were used for training and validation of the fuzzy systems, having 3700 data vectors each. The ranges of x1, x2 and y are [1072C, 1192C], [16.5 s, 20.5 s], and [1045C, 1125C] respectively.
- The Results section is very brief and does not really present any results. It only indicates how the error of the models will be evaluated, but the results are shown in the Discussion section. Both sections should be reorganized.
Response.
We modified this section, we included and described tables and figures in the result section, all this is in blue characters. We also make changes suggested by another reviewer, which are in red characters.
- In order to facilitate the reading and understanding of the tables and figures in which the authors present their results, they should be explained in greater detail and properly ordered, facilitating comparison. For example, Figures 6 to 9 appear consecutively without any explanatory text.
Response
We hope that with the reordered of the sections mentioned in the previous point may be better understood. We also included some extra statements such as:
Table 2 shows only the ‘%Bars ±20C’ performance index of the different varieties of fuzzy systems tested so far, since it allows a more straight forwardly comparison.
In order to allow a more detailed comparison, Table 3 shows the performance indices of the top four GBs, with 100% of ‘%Bars±20C’, ranked by mean error.
Scatter plots show predictions against measurements; hence the ideal prediction would be a unitary ramp.
The ideal prediction line is the one plotted in white squares.
Some other comments were added in the discussion section in blue characters, with the purpose of making it more comprehensive.
- Authors should compare their results with those of other researchers to improve the content of the Discussion section.
Response. Thank you for the suggestion. As mentioned, unfortunately we have not found yet works with which we could compare it directly

Reviewer 2 Report
STRONG POINTS
- The investigated problem is interesting both from theoretical and practical points of view.
- The selection of the methods tested for the estimation of finishing mill entry temperature is adequate and includes a satisfactory range of fuzzy-based approaches together with the combination of other techniques that try to overcome single methods weaknesses
- results are convincing and their analysis puts into evidence the main outcomes of this research
WEAK POINTS
- too many acronyms, some of them non necessary (i.e. SB, PF,..) that compromise the readability of the paper
- some background about the employed approaches must be provided, otherwise the understanding of the rationale of the research is precluded to part of the readers
- explain the difference bewteen type-1 and type-2 fuzzy systems
- section 2.2: the more complex approaches proposed in the paper are suitable to the problem and the sinergy between the parts works fine and is conceptually correct but the motivation behinh it should be explained in detail
- pag. 8 line 156: not readable
- figures 8 and 9: you clim that the performance of the PF models is poor at high and low temperatures but the plot (a cloud) puts into evidence that the models do not perform satisfactory in general
- Conclusions: worried about computational performance but this aspect should only affect the system during its training. I do not expect problems from the employed models in the simulation phase.
Author Response
Reviewer 2
Comments and Suggestions for Authors
Dear reviewer, we appreciate your time and your recommendations. The amendments to your observations are in green characters in the main text or in orange characters, if similar recommendations were done by another reviewer.
STRONG POINTS
- The investigated problem is interesting both from theoretical and practical points of view.
- The selection of the methods tested for the estimation of finishing mill entry temperature is adequate and includes a satisfactory range of fuzzy-based approaches together with the combination of other techniques that try to overcome single methods weaknesses
- results are convincing and their analysis puts into evidence the main outcomes of this research
WEAK POINTS
- too many acronyms, some of them non necessary (i.e. SB, PF,..) that compromise the readability of the paper
Response. We appreciate the observation. We reduced the number of acronyms to 9 and included a list of symbols and acronyms in Table 1. However, the table is in red characters since it was explicitly suggested by another reviewer.
- some background about the employed approaches must be provided, otherwise the understanding of the rationale of the research is precluded to part of the readers
Response. Thank you for the suggestion. We included two subsections in green characters in the methodology section, 2.2.1 and 2.2.2 for fuzzy systems, including type 2, and fuzzy c-means.
- explain the difference between type-1 and type-2 fuzzy systems
Response. Thank you for the observation. The difference is explained in the subsections included in the previous point.
- section 2.2: the more complex approaches proposed in the paper are suitable to the problem and the sinergy between the parts works fine and is conceptually correct but the motivation behinh it should be explained in detail
Response.
We appreciate the observation, we included the following statement in the introduction section in green characters:
In this work, HL type-2 Sugeno fuzzy systems, both, with empirical and FCM rule generation, will be developed and evaluated for scale breaker entry temperature prediction to take advantage of the merits of both techniques combined. Owing to the importance of scale breaker entry temperature for the finishing mill set-up, and hence, for bar head-end to meet requirements; the main purpose of this work is to explore the benefits of these two methods combined on scale-breaker temperature estimation, given the simplicity of Sugeno systems in relation to Mamdani and the powerfulness of type-2 fuzzy.
I think that was really our motivation. We also included some comments in green characters all over the document that may help to make it clearer.
- 8 line 156: not readable
Response
Thank you. We apologize for that. For a reason, in the journal template seems to be disabled the function to insert an equation, I had to write it as a text, unfortunately, in the section you suggested we have to include some equations, we had to make copy paste cause it was impractical to write them as a text, and some weird things happened. If the paper goes further, I will have to request assistance from the journal.
- figures 8 and 9: you clim that the performance of the PF models is poor at high and low temperatures but the plot (a cloud) puts into evidence that the models do not perform satisfactory in general
Response: Yes, you are right, we left it as “clearly, predictions have to be improved.”.
- Conclusions: worried about computational performance but this aspect should only affect the system during its training. I do not expect problems from the employed models in the simulation phase.
Response. You are right again, we removed from the conclusion section but we’ve written the following statement:
As mentioned, two out of four GB model with 100% of ‘%Bars±20C’ and the top two purely fuzzy systems are HL type-2 Sugeno, however the benefits brought to the purely fuzzy systems are marginal with respect to type-1 Mamdani. The ultimate goal is to implement the kind of systems presented here in the real world; if this would be the case and this work may be useful as a guideline, in order to make a choice due to the similar performance shown here, evaluation of the algorithms efficiency in terms of computing time, should be performed before the actual real hot strip mill implementation. It would depend on the computing platform used, as well as the particular algorithm implementation.

Reviewer 3 Report
An interesting work that utilizes computational intelligence techniques to predict temperature in hot rolling operation. The results are adequately presented and support the conclusions. The paper is well-organized and can be considered for a potential publication. However, appropriate revisions are needed before publication:
- The quality of Figure 1 is low and is difficult to read the symbols and letters. Please provide the same figure that all the important information is clearly marked (e.g. bigger font size or higher resolution). Please also mark the direction of the process.
- It is very interesting to see the evolution of a project and the phases that it been through, that are well described by the authors. However, a more extensive literature review is necessary in the “Introduction” section. The existing literature review is more a review of the author’s group publications and does not reflect the state of the art adequately. Please provide more references and more up to date (of the last 5-10 years) that support your claims and highlight the innovation content of your important work.
- Extensive editing required:
- Typo error. Both sections in line 84 and 92 are named 2.2. Change to 2.3 in the second case.
- The numbering of functions and rules is not correct. Line 112 is numbered as 1, line 143 as 5 instead of 2 and line 156 has no numbering.
- Check the alignment and font size to be consistent for all functions and their numbering
- Correct function errors (e.g. Function in line 143 has a frame around and in line 156 there is a square between the symbols)
- An abbreviation list is recommended at the beginning of the manuscript. There are many abbreviations in the document and it difficult even for an experienced reader to follow the text. An abbreviation list at a certain place will make everything easier instead of searching the meaning of each abbreviation in various places in the text.
- Figures 3-5 including the paragraphs where they are referenced and described according to my understanding are results of your model. However, they are located in the methodology section (Section 2) instead of the results section (Section 3). What is purpose of that?
Author Response
Dear reviewer, we appreciate your time and your recommendations. The amendments to your observations are in red characters in the main text or in orange characters, if similar recommendations were done by another reviewer.
An interesting work that utilizes computational intelligence techniques to predict temperature in hot rolling operation. The results are adequately presented and support the conclusions. The paper is well-organized and can be considered for a potential publication. However, appropriate revisions are needed before publication:
- The quality of Figure 1 is low and is difficult to read the symbols and letters. Please provide the same figure that all the important information is clearly marked (e.g. bigger font size or higher resolution). Please also mark the direction of the process.
Reponse.
Thank you very much for the observation, we improved it and added an arrow showing the rolling direction.
- It is very interesting to see the evolution of a project and the phases that it been through, that are well described by the authors. However, a more extensive literature review is necessary in the “Introduction” section. The existing literature review is more a review of the author’s group publications and does not reflect the state of the art adequately. Please provide more references and more up to date (of the last 5-10 years) that support your claims and highlight the innovation content of your important work.
Response.
Thank you again.
We removed four of our self-references, making indirect citations, and included eight new ones, only one of them is from 2009 and the rest are within the last 5 years. We organize them in three groups as follows:
- Unfortunately, we did not find works directly related to the ours, i.e. scale breaker entry temperature estimation, we included three research works in which temperature at different points of the hot rolling line is estimated by physical modeling, including scale breaker, however evaluated by the performance to estimate the finishing temperature. In fact, the only previous work related to the ours, that we have found, is the physical model used at the rolling mills with which we are comparing our work.
- We included two works with the aim of showing that estimation and control in industry applications via computational intelligence techniques is a research active area.
- We included three works in which some rolling variable are estimated by intelligent techniques in hot rolling to show as well that this in an active research area in hot rolling.
The changes are in orange color characters since another reviewer made a similar suggestion.
- Extensive editing required:
- Typo error. Both sections in line 84 and 92 are named 2.2. Change to 2.3 in the second case.
Response, we have corrected this, however new sections were added, we hope we got it right this time.
- The numbering of functions and rules is not correct. Line 112 is numbered as 1, line 143 as 5 instead of 2 and line 156 has no numbering.
Response, we have corrected this, however new equations were added.
- Check the alignment and font size to be consistent for all functions and their numbering
Response, Thank you. We apologize for that. For a reason, in the journal template seems to be disabled the function to insert an equation, I had to write it as a text, unfortunately, some sections had to be included along with some equations, we have to make copy paste cause it was impractical to write them as a text, and some weird things happened. If the paper goes further, I will have to request assistance from the journal.
- Correct function errors (e.g. Function in line 143 has a frame around and in line 156 there is a square between the symbols)
Please, see previous point.
- An abbreviation list is recommended at the beginning of the manuscript. There are many abbreviations in the document and it difficult even for an experienced reader to follow the text. An abbreviation list at a certain place will make everything easier instead of searching the meaning of each abbreviation in various places in the text.
Response. Thank you for the observation. We reduced the number of acronyms to 9 and included a list of symbols and acronyms in Table 1 in red characters.
- Figures 3-5 including the paragraphs where they are referenced and described according to my understanding are results of your model. However, they are located in the methodology section (Section 2) instead of the results section (Section 3). What is purpose of that?
Response. Thank you. these figures and their corresponding texts were relocated into the results section, they are in red characters.

Round 2
Reviewer 3 Report
Thank you for your effort to address to all comments.
The reply to all comments is convincing and the modifications to the manuscript are well organized.
Minor editing improvements must be made in some functions.
In my humble opinion I believe that the manuscript can be published in the present form.
Author Response
26 may 2020
Academic Editor Notes
Dear reviewer, we appreciate your time and your recommendations. The amendments to your observations have been addressed, we hope satisfactorily. Three little amendments were made to the main text and they are in red characters.
The manuscript reads well now.
However, I would like a very small number of correction added.
Response: Thank you
I would like to see an improved "y-axis" title for graphs 8-11. Your current description is too verbose and detailed for a graph axis, I think simply "Fuzzy-system predicted Temperature (C)" would be nicer.
Response: thank you for the observation, we have changed the y-axis labels in those figures.
Also, the equations are still not displaying properly on the whole. please check that all Eq's are embedded okay and convert to pdf okay.
Please remove the surrounding frames of Eq 3,4,5.
Please check carefully Eq 6 and 7, they are missing more than half the relevant symbols, often it is the integral symbol that has got corrupted.
Yet Eq 8, seems fine, so the template should work if used correctly.
Response: Thank you again. All the equations were edited, they the proper format now.
And I do not think that your current Eq 1 and 2 are really equations, they are logic statements, so I would rather see the numbering for them removed from the usual Eq. numbering.
Response: Their numbers were deleted, and all the equation numbers modified accordingly, however, it involved three little changes in the text, two in section 2.2.1, and another one in section 2.5, they are in red characters.
